# Preparation and Thermal Analysis of Blended Nanoaluminum/Fluorinated Polyether-Segmented Urethane Composites

**Chance Melvin Baxter [1], Jena McCollum [2]**  **and Scott Thomas Iacono [1,]***

[1] Department of Chemistry, Chemistry Research Center, Laboratories for Advanced Materials, United States Air Force Academy, Colorado Springs, Colorado, CO 80911, USA; chance.m.baxter@gmail.com

[2] Department of Mechanical and Aerospace Engineering, University of Colorado-Colorado Springs, Colorado Springs, Colorado, CO 80918, USA; jmccollu@uccs.edu

* Correspondence: scott.iacono@usafa.edu; Tel.: +1-719-333-6005

**Abstract:** The thermally induced reaction of aluminum fuel and a fluoropolymer oxidizer such as polytetrafluoroethylene (via C-F activation) has been a well-studied thermite event for slow-burning pyrolants among a multitude of energetic applications. Generally, most metallized thermoplastic fluoropolymers suffer from manufacturing limitations using common melt or solvent processing techniques due to the inherent low surface energy and high crystallinity of fluoropolymers. In this report, we prepared an energetic composite utilizing the versatility of urethane-based polymers and provide a comparative thermal characterization study. Specifically, a thermite formulation comprising of nanometer-sized aluminum (nAl) fuel coated with perfluoropolyether (PFPE) oxidizer was solvent-blended with either a polyethylene glycol (PEG) or PFPE-segmented urethane copolymer. Thermal data were collected with calorimetric and thermogravimetric techniques to determine glass transition temperature and decomposition temperature, which showed modest effects upon various loadings of PFPE-coated nAl in the urethane matrix. While our application focus was for energetics, this study also demonstrates the potential to expand the ability to broadly manufacture structural metallized composites to their consideration as coatings, foams, or fibers.

**Keywords:** urethanes; fluoropolymers; composites; aluminum

## 1. Introduction

The thermite reaction between aluminum and a fluoropolymer (e.g., polytetrafluoroethylene) has been well studied as a sustained burning pyrolant system for a multitude of specialized energetic [1,2], as well as broader commercial, applications [3]. However, most of such composite formulations lack significant structural integrity to be used for practical applications. As such, we have been investigating the utility of preparing metastable matrix composites utilizing a core/shell formulation approach comprising of nanometer-sized aluminum (nAl) as the fuel (the core) and a physio-adsorbed fluoropolymer coating, specifically a class of fluoropolymers called perfluoropolyethers (PFPEs) [4] as the oxidizer (the shell). These energetic composites have included postmachinable epoxy-based thermoset molds [5,6], electrospun polystyrene microfibers [7], and melt-processed fluoropolymer thermoplastics [8]. Overall, these systems are limited to the amount of nAl loading due to the poor metal interface with an organic-based matrix. Consequently, this results in incompletely crosslinked/cured materials, melt and solvent processing challenges, and/or deleterious effects on mechanical properties.

In this work, we expanded our work on PFPE-formulated energetic composites by studying the versatile class of polyurethanes as host matrices [9], specifically a PFPE-segmented urethane



copolymer that demonstrated excellent compatibility with PFPE-pretreated nAl particles at high weight percent loadings. By way of comparison, a nonfluorinated, structurally similar polyethylene glycol (PEG) urethane copolymer was prepared, which suffered from moisture retention but surprisingly retained PFPE-pretreated nAl particles at high loadings. This work affords comprehensive thermal properties using calorimetric and gravimetric means of these composite systems formulated by solvent blending, which is useful for urethane-based coatings, blown foams, and the fiber industry and has potential opportunity in emerging solvent-cast direct-write fabrication technologies for microstructure components.

## 2. Materials and Methods

### 2.1. Materials

All reagents were used as received. Perfluoropolyether (PFPE) Fomblin-Y (LVAC 25/6, avg mol wt 3300 g/mol), polyethylene glycol (PEG, average molecular weight 3000 g/mol), isopherone diisocyanate (IPDI), and tin(II) 2-ethylhexanoate were purchased from Sigma Aldrich. Reagent grade tetrahydrofuran (THF) and perfluoropolyether (PFPE) diol, tradename Fluorolink D10, was purchased from Alfa Aesar. Nanometer-sized aluminum (nAl) powder was obtained from the US Army Armament Research, Development, and Engineering Center (ARDEC) and had an average particle size distribution of 80 nm, as determined by transmission electron microscopy by the supplier. The manufacturer found the aluminum to be ca. 70% active, as determined by thermogravimetric analysis by measuring the mass gain due to oxidation.

### 2.2. Methods

Attenuated total reflectance Fourier transform infrared (ATR-FTIR) spectra were collected using a Thermo Nicolet FTIR spectrometer iS10 (Thermo Fisher Scientific, Waltham, MA, USA). Differential scanning calorimetry (DSC) was performed on a TA Auto Q20 Instrument in nitrogen (New Castle, DE, USA). Samples (ca. 5 mg) were sealed in an aluminum hermetic pan with an empty sealed hermetic pan serving as the reference. Thermal transitions were reported on the third heating cycle. Samples were heated/cooled at a rate of 5 °C/min. Thermal gravimetric analysis (TGA) was performed on a TA Q500 instrument at a scan rate of 5 °C/min in nitrogen. Samples (5–10 mg) were measured with a platinum crucible and heated from room temperature to 900 °C. TA Universal Analysis 2000 graphical software was used to determine the glass transition ($T_g$) and decomposition temperatures ($T_d$), along with remaining mass balance (nAl bal)/char yields (%). All data were shown to be repeatable to within ±5% of determined values.

### 2.3. General Procedure for Synthesis of Fluorinated (F) or Nonfluorinated (NF) Urethane Copolymers

A master batch (typically a 10-g basis) of the two polyether-segmented polyurethane systems were prepared via the same reaction procedure as follows. The copolymers (Scheme 1) were prepared by mixing a stoichiometric equivalent of the respective diol monomer (PEG or Fluorolink D10) and the IPDI in a disposable glass vial. Tin(II) 2-ethylhexanoate (~1 wt %) was added to catalyze the polymerization with constant stirring for 5 min. The urethane copolymer was ready for preparing solvent-blended composites (see subsequent Section 2.4) after allowing the formulation to set for 24 h at room temperature, which afforded a transparent and opaque solid for the nonfluorinated PEG-based (denoted "**NF**") and fluorinated Fluorolink D10 (denoted "**F**") copolymers, respectively.

**Scheme 1.** Synthesis of fluorinated (**F**) or nonfluorinated (**NF**) polyether-segmented urethane copolymers.

## 2.4. nAl/Fomblin-Y Blend Preparation

Following a previously published procedure [5], a master batch of 30 wt % nAl in Fomblin-Y was prepared in a glovebox under a nitrogen atmosphere by weighing components directly into a glass screw cap vial and manually mixing them with a spatula for 5 min. The vials were then capped and the blends removed from the glove box for composite preparation.

## 2.5. General Procedure for nAl/PFPE-Blended Fluorinated (F) or Nonfluorinated (NF) Urethane Copolymer Composites

A minimal amount of THF was added to dissolve either the fluorinated PFPE (denoted "**F**") or nonfluorinated PEG (denoted "**NF**") polyether-segmented urethane copolymer using a recorded dry weight for processing with the nAl/Fomblin-Y blends (Scheme 2). Once the polymer was dissolved, the samples were sectioned into six equivalent amounts by mass. The preformulated nAl/Fomblin-Y blend was added to the samples in 0%, 10%, 20%, 30%, and 50% by nAl weight and manually stirred for 5 min. Thin films of the composite samples were prepared by drop casting the formulation onto glass slides, and the solvent was allowed to evaporate after 24 h upon air drying. The samples were then analyzed using DSC and TGA techniques to record the thermal transitions.

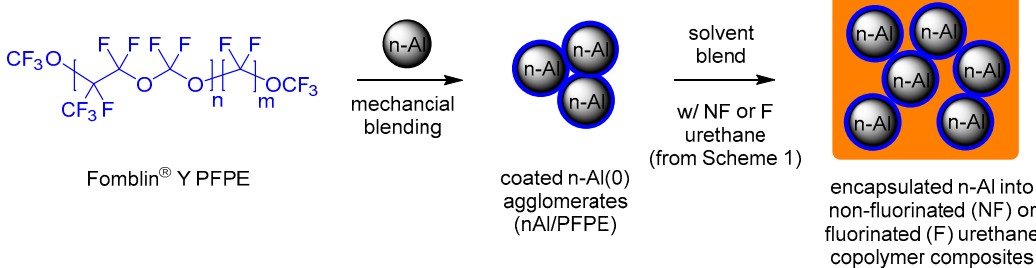

**Scheme 2.** Solvent blending nanometer-sized aluminum (nAl)/perfluoropolyether (PFPE) blends with fluorinated (**F**) or nonfluorinated (**NF**) polyether-segmented urethane copolymers.

## 3. Results and Discussion

### 3.1. Preparation of nAl/PFPE-Blended Fluorinated (F) or Nonfluorinated (NF) Urethane Copolymer Composites

Two distinct segmented urethane copolymers were prepared utilizing either a fluorinated or nonfluorinated polyether diol (Scheme 1). Specifically, perfluoropolyether (PFPE) diol or polyethylene glycol (PEG) diol was reacted with catalytic tin(II) 2-ethylhexanoate to afford fluorinated (**F**) or nonfluorinated (**NF**) urethane copolymers, respectively. ATR-IR analysis of both linear copolymers confirmed the complete conversion of the alcohol (O–H) stretch at 3300 cm$^{-1}$ and the isocyanate (–N=C=O) asymmetric and symmetric stretches at 2240 cm$^{-1}$ and 1386 cm$^{-1}$, respectively, to the expected carbamate linkage stretches comprised of 1700 cm$^{-1}$ (C=O), 1513 cm$^{-1}$ (N–H δ), and 1238 cm$^{-1}$ (asymmetric N–CO–O). No additional characterization was deemed necessary given the established chemistry of similar PFPE/IPDI-segmented urethane copolymers systems that had previously been investigated [10].

Both **NF** and **F** copolymers were prepared in 10-g individual batches and exhibited excellent solubility in common organic solvents such as tetrahydrofuran (THF). Utilizing a minimal amount of THF, free-standing 1–2-mm-thick transparent or opaque films from **NF** and **F** copolymers, respectively, were produced by drop or blade casting the formulation on glass microscope slides and were allowed to dry for 24 h. The resulting homogeneous opaque films produced from the **F** copolymer indicated an expected phase separation of the fluorine-rich segments from the nonfluorinated isopherone moiety. Fomblin-YPFPE (PFPE oligomer end-capped with CF$_3$) surface-treated nAl (30 wt %) was solvent-blended with either **NF** or **F** urethane copolymers using a minimal amount of THF at various weight percent loadings (Scheme 2). The specific use of 30 wt % nAl PFPE-treated particles in this study was a result of previous thermal studies that indicated the stoichiometrically optimized formulation for energetic release based on calorimetry analysis [3]. The samples used for subsequent thermal analysis were prepared by drop or blade casting onto microscope slides and were allowed to dry for 24 h in open air. The nAl/PFPE-loaded **F** and **NF** films produced dull gray, free-standing films with no appearance of phase separation for loadings to 50 wt %. Weight percent loadings higher than 50% yielded phase-separated films for both **F-** and **NF**-segmented copolymer composites. Attempts to produce control films with comparable weight percent loadings of untreated nAl (no PFPE) failed to produce homogeneous films. This was the first indication that an organic coated barrier was required for improving the nAl interface with an urethane copolymer matrix.

### 3.2. Thermal Analysis of nAl/PFPE-Blended Fluorinated (F) or Nonfluorinated (NF) Urethane Copolymer Composites

The data collected provided thermal transitions of composites of nonfluorinated/fluorinated polyether-segmented urethane copolymers solvent-blended with preformulated nanometer-sized aluminum/perfluoropolyether (nAl/PFPE) blends. The effects upon the temperature on these composites were measured by differential scanning calorimetry (DSC) and thermogravimetric analysis (TGA), and selected properties for the focus of this study are summarized in Table 1.

**Table 1.** Consolidated table of selected composite thermal analysis data of nAl/PFPE solvent-blended into nonfluorinated (**NF**) or fluorinated (**F**) polyether-segmented urethane copolymers.

| nAl/PFPE Loading (wt %) | $T_g$ (°C) | $T_d$ (°C) | Char Yield (%) |
|---|---|---|---|
| 0 **NF** | 32 | 283 | 1 |
| 10 **NF** | 25 | 276 | 2 |
| 20 **NF** | 25 | 278 | 8 |
| 30 **NF** | 23 | 274 | 9 |
| 50 **NF** | 27 | 272 | 13 |
| 0 **F** | 6 | 249 | 1 |
| 10 **F** | 5 | 245 | 11 |
| 20 **F** | 5 | 247 | 14 |
| 30 **F** | −1 | 243 | 16 |
| 50 **F** | −1 | 250 | 21 |

The measured glass transition ($T_g$) by DSC showed the unfilled **NF** copolymer 26 °C higher than the **F** copolymer, as shown in Figure 1. Specifically, the PFPE system studied resulted in a $T_g$ below room temperature of 6 °C, affording a true elastomeric urethane. This was expected given that the phase separation from the measured response of the IPDI "hard" domains caused a plasticizing effect induced by the PFPE-rich "soft" domains. It is important to note the $T_g$ of the PFPE segments was measured using DSC at −116 °C [11], but our DSC capabilities were limited to −90 °C. Nonetheless, the bulk properties of the urethane system, specifically the mechanical integrity, were captured as a result of the IPDI hard domain glass transition response. For the composites, the DSC analysis of the increasing weight percent (wt %) of the nAl/PFPE blend (comprised of 30 wt % nAl in PFPE) in either the **NF** or **F** urethane copolymer afforded the measured $T_g$, as shown in Figure 2. The overall treads for loadings up to 50 wt % of the nAl/PFPE blends in either the **NF** or **F** urethane copolymers resulted in a modest plasticizing effect observed by a slight lowering of the $T_g$.

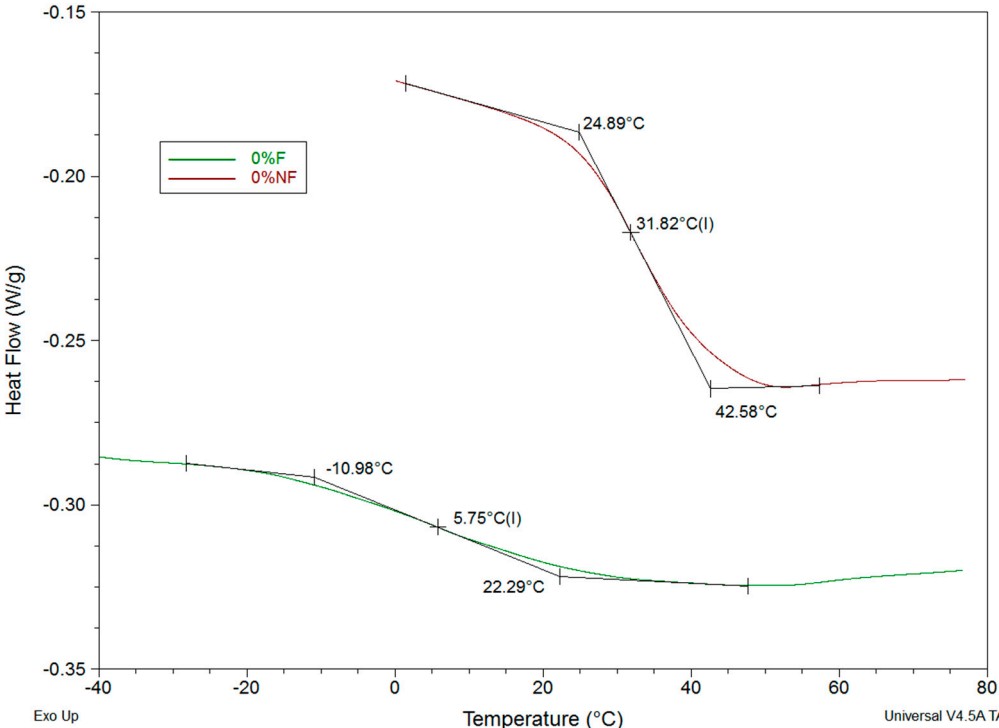

**Figure 1.** Differential scanning calorimetry (DSC) overlay of fluorinated and nonfluorinated polyether-segmented urethane copolymers with no nAl/PFPE loading.

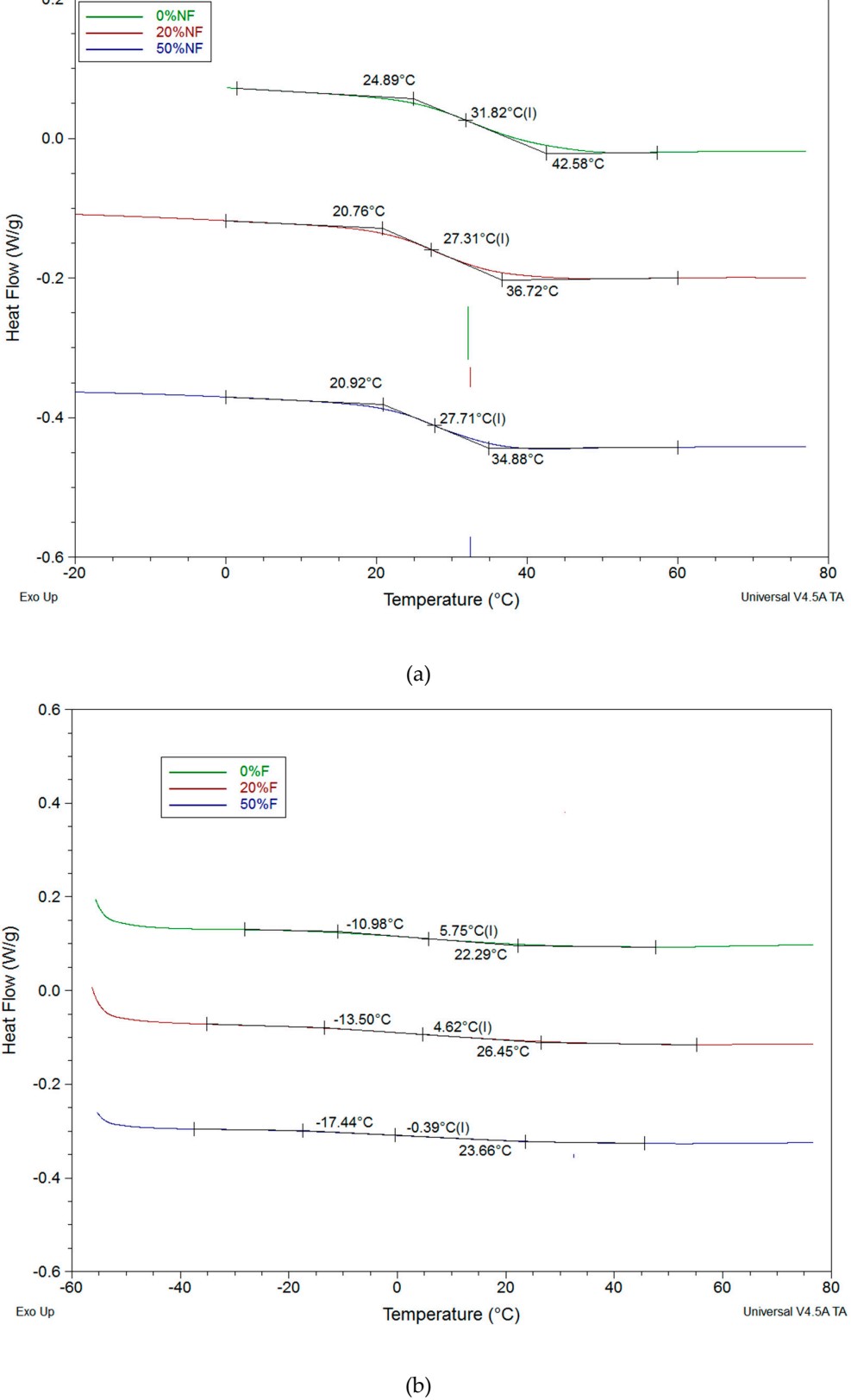

(a)

(b)

**Figure 2.** DSC overlay of nonfluorinated polyether-segmented polymer urethane copolymers with selected nAl/PFPE loadings (**a**). DSC overlay of fluorinated polyether-segmented urethane copolymers with selected nAl/PFPE loadings (**b**).

TGA analysis measured the onset of polymer decomposition ($T_d$) and resulting char yield (%) recorded at 900 °C in nitrogen, as shown in Figure 2 for the unfilled **NF** and **F** urethane copolymer and in Figure 3 for the increased loading of selected nAl/PFPE-blended composites. As shown Figure 1, both the unfilled **NF** and **F** copolymers exhibited negligible char yield, which is expected for aliphatic-rich urethane-based systems. The TGA of the PEG-based urethane copolymer (**NF**) showed an initial 7–8 wt % loss onset at 85 °C, which is consistent with moisture evaporation of such hydrophobic urethanes [12]. The same observation was revealed even when attempting to mitigate with oven drying immediately before TGA analysis. On the other hand, hydrophobic PFPE-enriched urethane copolymer **F** demonstrated a flat thermogravimetric trace until onset of decomposition at 249 °C, which was 34 °C lower than the PEG **F** urethane copolymer. This trend was consistent with nAl/PFPE solvent-blended composites of the **NF** and **F** urethane copolymers, as shown in Figure 4. Introducing fluorine to many polymer systems inherently increases thermal stability due to the thermodynamically stronger C–F bonds (544 kJ/mol), compared to C–H bonds (414 kJ/mol). However, in the PFPE-segmented urethane copolymer system, a weaker carbamate linkage existed as a result of the incorporation of a deactivated, electron-withdrawing PFPE diol compared to the PEG diol [13]. Overall, the char yields increased with increased loading of nAl/PFPE, as calculated from a metal-only (Al) content basis. Of additional statistical significance, the composites formulated from the **F** copolymer demonstrated consistently higher char yields (by up to 8 wt %) than the **NF** copolymer. This was observed in a prior report, where the available fluorine from the PFPE segment oxidized the Al surface, resulting in a slightly higher uptake in weight as the copolymer underwent decomposition in an inert environment [14]. In Figure 4, it should be noted that samples with nAl content, weight percent uptake started occurring at 550 °C, which was the result of the oxidation of Al to $Al_2O_3$ from trace (ppm) $O_2$ present in the $N_2$ carrier gas.

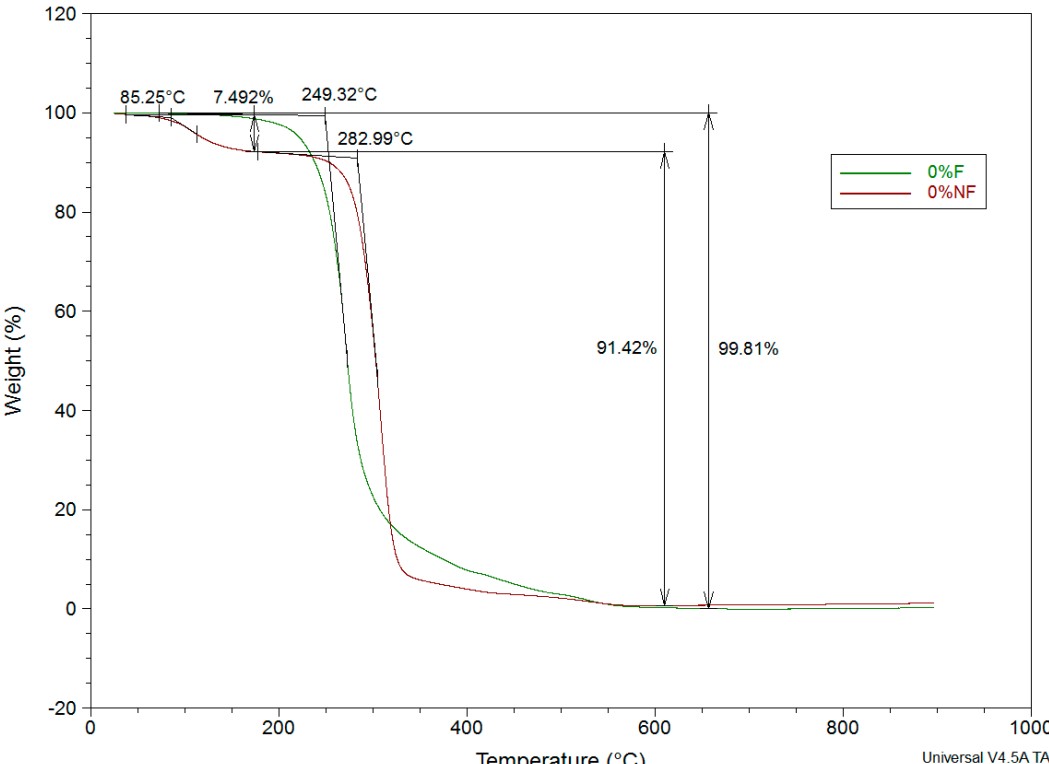

**Figure 3.** Thermogravimetric analysis (TGA) overlay of fluorinated and nonfluorinated polyether-segmented urethane copolymers with no nAl/PFPE loading.

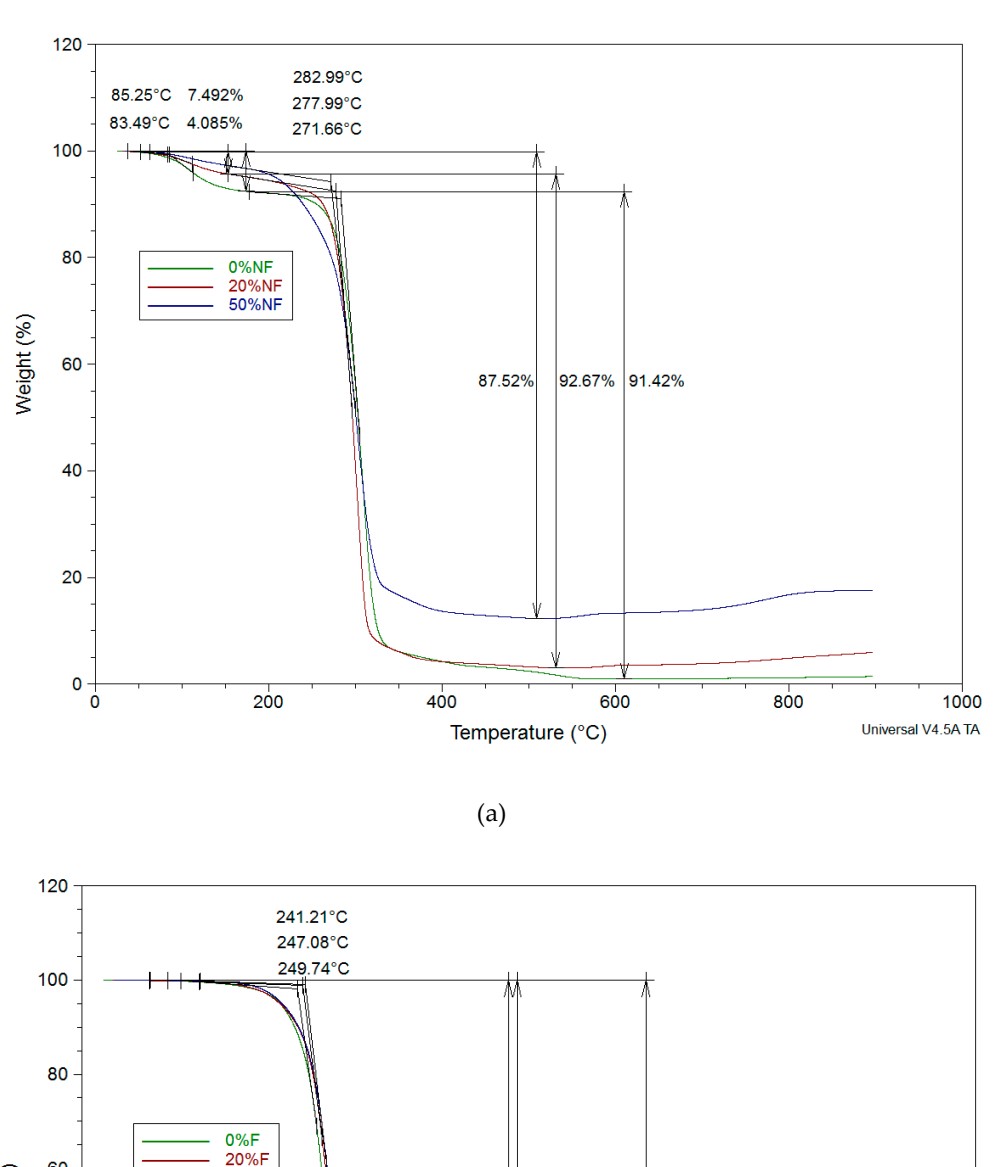

(a)

(b)

**Figure 4.** TGA overlay of nonfluorinated polyether-segmented urethane copolymers with selected nAl/PFPE loadings (**a**). TGA overlay of fluorinated polyether-segmented urethane copolymers with selected nAl/PFPE loadings (**b**).

## 4. Conclusions

In summary, we reported a comparative preparation and thermal analysis of aliphatic PEG (nonfluorinated) and PFPE (fluorinated) segmented urethane matrices solvent-blended with PFPE-treated nAl. Formulations of these composites demonstrated excellent compatibility in both matrices, with nAl/PFPE blends at high loadings of up to 50 wt % (resulting in 15 wt % Al content), affording free-standing, drop- or blade-cast films from solvent. Without pretreatment of nAl with PFPE, only phase-separated composite films were produced with poor mechanical integrity. In addition, the thermal properties validated the compatibilization in the amorphous domains of the urethane matrix by demonstrating negligible deviations in glass transition temperature. This suggests that PFPE pretreatment of nanometer-sized Al could be utilized by other urethane copolymers as versatile host matrices for metal particles (potentially not limited to Al) and be utilized as coating, foams, or fibers in commercial applications. For our continued interest in thermite-infused composites, this study necessitates the additional elucidation of quantitative thermal outputs and burn efficiency of the prepared urethane copolymer matrices. As such, further pre- and post-analysis using scanning or electron emission microscopy is planned in order to observe the degrees of dispersion within the composite materials and their correlation to open air burn studies. Finally, although the PEG-segmented urethane copolymer demonstrated slightly higher thermal decomposition compared to the PFPE-segmented urethane copolymer, it suffered from moisture absorption, which is detrimental in such energetic applications.

**Author Contributions:** S.T.I. designed and coordinated the research work, and C.M.B. and J.M. performed all the experiments.

**Funding:** This research was funded by the Air Force Office of Scientific Research (AFOSR) under the memorandum of agreement between the United States Air Force Academy and the Chemistry Research Center, Laboratories for Advanced Materials.

**Conflicts of Interest:** The authors declare no conflict of interest. The funders had no role in the design of the study; in the collection, analyses, or interpretation of data; in the writing of the manuscript; or in the decision to publish the results.

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
