# Peer review of "Preparation and Thermal Analysis of Blended Nanoaluminum/Fluorinated Polyether-Segmented Urethane Composites"

_jcs, doi:10.3390/jcs3010025_

Round 1
Reviewer 1 Report
The authors presented a comparative preparation and thermal analysis of aliphatic PEG and PFPE segmented urethane matrices solvent blended with PFPE treated nAl.
1. L174-L178 : "Introducing fluorine to many polymer systems inherently increases thermal stability due to the thermodynamically stronger C–F bonds compared with C–H bonds."; Indicate the values of C-F bonding energy, and also C-H bonding energy.
"However, in the PFPE segmented urethane copolymer system, a weaker carbamate linkage exists as a result of the more acidic PFPE diol compare with the PEG diol [13]."; Explain more carefully with giving a value of carbamate linkage energy.
2. As for Figure 2 and Figure 4, please explain Figure 2 (a),(b) and Figure 4(a), (b), not in (top) and (bottom).
Author Response
Point 1: Adjustments made in the tracked changes in the manuscript per reviewer's suggestion
Point 2: Seems minor, we kept the format since this is consistent with similar MDPI Figure entries and we feel it's really easier to read for this type of data entry since the figures are crowded with numerical labels.
Reviewer 2 Report
In reviewed manuscript authors present results of their study on thermal properties of advanced composites. I find this, short, paper interesting, well prepared and it perfectly fits into scope of the journal. I suggest to accept it after just minor revision, which is:
1. Authors should use SI units through manuscript, and in SI, the unit of the temperature is Kelvin, not Celsius.
2. Authors should not use comas in numbers (like do for example on page 2, line 55).
3. Uncertainty should be discussed, and uncertainty bars should be presented on the plots.
Author Response
Point 1: We re-reviewed the manuscript for units and everything is reported in SI units.
Point 2: Corrections made per reviewer's comments
Point 3: We're not sure how to report error bars given our data. As reported as figures, these are singular representations of data for illustrative purposes. However, this is an important comment, and per recommendation, we have placed a statement reported in the experimental that 'repeated results are withing +/- 5% error.'
Reviewer 3 Report
The authors submitted the article entitled “Preparation and Thermal Analysis of Blended Nanoaluminum/Fluorinated Polyether-Segmented Urethane Composites”. I would recommend that the paper could be published elsewhere. My main comments and questions are as follows:
1. Overall, the motivation is fine but the introduction should be improved.
2. The only discussion of thermal properties cannot match the scope of the journal.
3. The data are scientifically insufficient and insignificant. The authors should provide more data, such as NMR, GPC, DSC, and so on for basic characterization in the main text.
4. Lists of sample characterization are needed.
Author Response
We are attempting to take the reviewer's comments/suggestions seriously, but we think there is a lot of ambiguity in this review. Below is our best attempt to address each point.
Point 1: Without more clarification, the comment is vague, we kept the Introduction the same since it suitably covers/cites the field pertaining to this work.
Point 2: We are not sure how to respond to this. This is really an Editor decision if our submission is outside of the journal's scope, but when considering this journal, our data shares the importance of surfaces/interfaces with the focus of thermal analysis which is within the scope of this journal.
Point 3: We also not sure how to respond to 'insufficient' and 'insignificant' data. This seems contradictory and lacks specificity. The focus of this work is clearly, 'thermal' analysis of the studied composite system (as pointed out in the Abstract and Introduction). We have provided a significant and sufficient amount of data that would be normally acceptable for the 'art' of this specialized field.
The reviewer is asking us to provide DSC which is the focus of this work. I'm not sure the reviewer even understands what he/she is asking for or has read the manuscript. Also, asking for NMR and GPC is inconsistent with formulating composites. We are concerned the reviewer may be outside his/her field of expertise to provide an objective and adequate review.
Point 4: We are not clear, again, on what the reviewer is asking. Lists of data are inconsistent with the flow of this work and data presented.
Round 2
Reviewer 3 Report
It is well-revised now.